# *Stephania tetrandra* and Its Active Compound Coclaurine Sensitize NSCLC Cells to Cisplatin through EFHD2 Inhibition

**DOI:** 10.3390/ph17101356

**Published:** 2024-10-11

**Authors:** Shu-Yu Hu, Tsai-Hui Lin, Chung-Yu Chen, Yu-Hao He, Wei-Chien Huang, Ching-Yun Hsieh, Ya-Huey Chen, Wei-Chao Chang

**Affiliations:** 1Graduate Institute of Biomedical Sciences, China Medical University, Taichung 404333, Taiwan; estherhu9205@gmail.com (S.-Y.H.); antuzyh@gmail.com (Y.-H.H.); whuang@mail.cmu.edu.tw (W.-C.H.); 2Department of Chinese Medicine, China Medical University Hospital, Taichung 404327, Taiwan; 010178@tool.caaumed.org.tw; 3Research Center for Cancer Biology, China Medical University, Taichung 406040, Taiwan; okada8@gmail.com; 4Center for Molecular Medicine, China Medical University Hospital, Taichung 406040, Taiwan; 5Program for Cancer Biology and Drug Discovery, China Medical University, Taichung 404333, Taiwan; 6School of Pharmacy, China Medical University, Taichung 404333, Taiwan; 7Division of Hematology and Oncology, Department of internal medicine, China Medical University Hospital, Taichung 404327, Taiwan; tsienkoala@yahoo.com.tw

**Keywords:** NSCLC, cisplatin, EFHD2, *Stephania tetrandra*, coclaurine, FOXG1

## Abstract

Background: Adjuvant chemotherapy, particularly cisplatin, is recommended for non-small cell lung carcinoma (NSCLC) patients at high risk of recurrence. EF-hand domain-containing protein D2 (EFHD2) has been recently shown to increase cisplatin resistance and is significantly associated with recurrence in early-stage NSCLC patients. Natural products, commonly used as phytonutrients, are also recognized for their potential as pharmaceutical anticancer agents. Result: In this study, a range of Chinese herbs known for their antitumor or chemotherapy-enhancing properties were evaluated for their ability to inhibit EFHD2 expression in NSCLC cells. Among the herbs tested, *Stephania tetrandra* (*S. tetrandra*) exhibited the highest efficacy in inhibiting EFHD2 and sensitizing cells to cisplatin. Through LC-MS identification and functional assays, coclaurine was identified as a key molecule in *S. tetrandra* responsible for EFHD2 inhibition. Coclaurine not only downregulated EFHD2-related NOX4-ABCC1 signaling and enhanced cisplatin sensitivity, but also suppressed the stemness and metastatic properties of NSCLC cells. Mechanistically, coclaurine disrupted the interaction between the transcription factor FOXG1 and the EFHD2 promoter, leading to a reduction in EFHD2 transcription. Silencing FOXG1 further inhibited EFHD2 expression and sensitized NSCLC cells to cisplatin. Conclusions: *S. tetrandra* and its active compound coclaurine may serve as effective adjuvant therapies to improve cisplatin efficacy in the treatment of NSCLC.

## 1. Introduction

Lung cancer is the leading cause of cancer-related incidence and death worldwide [1]. Non-small cell lung cancer (NSCLC), which accounts for 85% of newly diagnosed cases, is the most common type of lung cancer [2]. Surgical resection remains the standard treatment for patients with early-stage (stage I and II) NSCLC [3]. However, 30–55% of these patients relapse within five years of surgical resection, with recurrence primarily associated with postsurgical mortality in NSCLC patients [4]. The high-risk clinical features for NSCLC recurrence include poorly differentiated tumors, tumors larger than 4 cm, vascular invasion, visceral pleural involvement, and unknown lymph node status [5]. Adjuvant chemotherapy, particularly with cisplatin, has been recommended to reduce the risk of recurrence by eliminating remaining tumor cells, but it results in only a 4% increase in 5-year survival compared to patients who do not receive treatment [6].

EF-hand domain-containing protein 2 (EFHD2) is a conserved calcium-binding protein with F-actin bundling ability [7]. Apart from neurons, it has recently been found to be highly expressed in normal intestinal tissues, where it plays a role in protecting the intestine from inflammation [8]. EFHD2 enhances invadopodia formation by modulating actin dynamics, thereby increasing cancer invasion and metastasis [9]. Our previous study demonstrated that EFHD2 promotes epithelial-to-mesenchymal transition (EMT) in NSCLC cells and is significantly associated with postsurgical recurrence in patients with stage I NSCLC [10], suggesting EFHD2 as a molecular marker for predicting recurrence. Additionally, EFHD2 activates the NOX4-ROS-ABCC1 signaling pathway to reduce intracellular cisplatin levels, thereby conferring cisplatin resistance on NSCLC cells [11]. Consequently, EFHD2-targeting strategies may offer an opportunity to enhance responsiveness of NSCLC patients to adjuvant chemotherapy.

Specific inhibitors against EFHD2 are currently unavailable. Ibuprofen, a non-steroidal anti-inflammatory drug (NSAID), has been shown to downregulate EFHD2 by promoting its proteasomal and lysosomal degradation [11]. NSAIDs have been tested for their ability to improve adjuvant chemotherapy over the past decade [12,13]. The administration of NSAIDs after surgical operation is associated with longer overall and progression-free survival in patients with NSCLC [14]. However, the long-term use of high-dose NSAIDs can potentially cause severe adverse effects, including gastrointestinal damage, kidney dysfunction, hematological abnormalities, and allergic reactions [15]. Therefore, despite the substantial *efficacy benefit* of adding NSAIDs, the long-term use of ibuprofen as a sensitizer for adjuvant chemotherapy remains a concern.

Herbal remedies, with a history spanning thousands of years in Asian societies, have seen a rapid global expansion in use over the past three decades [16]. Although certain medicinal plants naturally contain toxic substances that can cause adverse reactions if used inappropriately, herbal products are well-recognized for long-term therapy and as phytonutrients in healthy supplements [17]. Historically, natural compounds with diverse anticancer activities have been viewed as a plentiful source of new anticancer agents. To date, more than 60% of pharmaceutical anticancer agents have been derived from natural compounds [18]. Additionally, natural compounds exhibit the potential to attenuate resistance against cancer chemotherapy, and numerous clinical trials have been conducted to evaluate the sensitizing effects of natural compounds on various types of cancers, including lung cancer [19].

*Stephania tetrandra S. Moore* (*S. tetrandra*) is widely distributed in the tropical and subtropical regions of Asia and Africa. The root of *S. tetrandra*, known in traditional Chinese medicine as Fang Ji, is commonly used for its diuretic, antimicrobial, anti-inflammatory, antirheumatic, and neuroprotective properties to treat a range of illnesses, including rheumatism, arthralgia, edema, beriberi, and eczema [20]. To date, a total of 67 alkaloids and 5 non-alkaloids have been identified from the root of *S. tetrandra* [21]. The pharmacological efficacy has been largely attributed to its rich content of bisbenzylisoquinoline alkaloids, mainly cyclanoline, fangchinoline, and tetrandrine. Cyclanoline is composed of two benzylisoquinoline moieties connected by a berberine bridge [22]. Its structure includes hydroxyl and methoxy groups, contributing to its solubility and reactivity. The compound shows anti-inflammatory and antimicrobial activities, and its hydrophobic nature aids its interaction with cellular membranes [23]. Both fangchinoline and tetrandrine have a similar bisbenzylisoquinoline structure with ether-linked moieties, except that fangchinoline has a hydroxyl (-OH) group, while tetrandrine has a methoxy (-OCH_3_) group at C7 [24]. Similar to cyclanoline, its structural features, such as multiple aromatic rings and methoxy groups, facilitate its lipophilicity and interaction with membrane proteins. In addition to the properties of anticancer, anti-inflammatory, and cardiovascular actions, the antiviral effects of fangchinoline have recently garnered widespread attention [25,26]. Tetrandrine is the most extensively and thoroughly studied molecule in the alkaloid components isolated from *S. tetrandra* [27]. It is a calcium channel blocker and is well-characterized for its anticancer, anti-inflammatory, and immunosuppressive properties [28]. Its chemical characteristics, such as hydrophobicity, aromaticity, and functional group positioning, allow for effective action against hepatocellular carcinoma, as indicated by the quantitative structure-activity relationship (QSAR) model [29].

In the present study, we hypothesized that herbal extracts could sensitize NSCLC cells to cisplatin through EFHD2 inhibition. To test this hypothesis, a series of Chinese herbs reported to have antitumor or chemotherapy-supporting properties were evaluated for their ability to inhibit EFHD2 expression in NSCLC cells. Among the herbs tested, *S. tetrandra* exhibited the highest efficacy in inhibiting EFHD2 expression in NSCLC cells. Through LC-MS identification and functional assays, coclaurine was characterized as a vital molecule in *S. tetrandra* responsible for EFHD2 inhibition. Consequently, we investigated the underlying mechanisms of coclaurine in EFHD2 inhibition and evaluated its effects on sensitizing NSCLC cells to cisplatin.

## 2. Results

### 2.1. Aqueous Extracts of S. tetrandra Inhibit EFHD2 Expression

To identify potential compounds for EFHD2 inhibition, we acquired testing materials by performing aqueous extraction on a series of Chinese herbs reported to have antitumor or chemotherapy-supporting properties (Table 1). To enhance extraction efficiency, the herbal raw materials were cut into small pieces (length < 1 cm) and extracted with distilled water (2 g herbs in 25 mL H_2_O) with stirring at 170 °C for 30 min. After high-speed centrifugation to remove precipitates and residues, the suspensions were filtered using a 0.22 μm pore size membrane (#6534; Sartorius, Göttingen, Germany). The concentration of each extract was determined by speed-vacuum drying and weighting (Table 1).

Safety and suitable quality are essential issues in the use of herbal remedies for public healthcare [16]. To this end, we tested the minimal dose of herbal extracts (IC90 for the NSCLC cell line H1299 in this study) for EFHD2 inhibition to reduce adverse effects, such as herb-drug interactions, which may alter the systemic bioavailability and pharmacokinetics of these drugs. The IC90 dose of each herbal extract was determined by MTT assay (Figure 1A), and the corresponding dilution folds of the herbal extracts are listed in Table 1. The IC90 dose was further used to evaluate the effect of herbal extracts on sensitizing H1299 to cisplatin by MTT assay. Both *S. tetrandra* and *Smilax china* (*S. china*) significantly increased cisplatin efficacy with synergy (Figure 1B). Additionally, *S. tetrandra* and *S. china* showed potent inhibition of EFHD2 in H1299 cells using Western blot assay (Figure 1C). Because *S. tetrandra* exhibited greater effectiveness in suppressing EFHD2 compared to *S. china*, we aimed to explore the characteristics of *S. tetrandra* in cisplatin sensitization in this study.

### 2.2. S. tetrandra Decreases Migration and Invasion of NSCLC Cells

Western blot validation revealed that *S. tetrandra* suppresses EFHD2 expression in a dose- and time-dependent manner (Figure 2A,B). Notably, the proliferation of the normal human bronchial epithelium cell line BEAS-2B was unaffected by *S. tetrandra* (Figure 2C), indicating no significant toxicity to normal lung cells under *S. tetrandra* treatment. Our previous study demonstrated that EFHD2 impacts the migration and invasion abilities of NSCLC cells [10]. Therefore, we verified whether *S. tetrandra* affects these metastatic properties. As expected, *S. tetrandra* significantly attenuated the cell migration and invasion of H1299 cells in a dose-dependent manner (Figure 2D,E).

### 2.3. Purification of the Key Components in S. tetrandra Responsible for EFHD2 Inhibition

To gain insight into the critical molecules of *S. tetrandra* responsible for EFHD2 inhibition, reversed-phase *high-performance liquid chromatography* (RP-*HPLC*, C18 column; Agilent 1260 Infinity II; Santa Clara, CA, USA) was conducted to analyze the component profile and purify the major fractions for identification. The HPLC signal pattern was primarily divided into four fractions, F1–F4 (Figure 3A). According to previously published reports, fractions F2–F4 represent the standard pattern of *S. tetrandra*, mainly containing the active components cyclanoline, tetrandrine, and fangchinoline [30,31]. The relative content of the major peaks F2–F4 was calculated by the ratio of peak area of each phytochemical to the total peak areas in the chromatogram monitored at 280 nm (Figure 3B). These major fractions, F1–F4, were collected and dried to remove solvents by a vacuum centrifugation concentrator, then reconstructed with the same volume of distilled water. Western blot validation indicated that fraction F1 exhibits the highest potential for EFHD2 inhibition and also suppresses the EFHD2-related signaling molecules ATP-binding cassette subfamily C member 1 (ABCC1) and NADPH oxidase 4 (NOX4) [11] (Figure 3C). To further identify the effective component in F1, two major peaks, f1-1 and f1-2, were purified for activity examination. Western blot revealed that the f1-2 fraction shows predominant inhibition of EFHD2, ABCC1, and NOX4 in a dose-dependent manner (Figure 3D).

### 2.4. Coclaurine Is a Critical Compound of f1-2 Responsible for EFHD2 Inhibition

To decipher the chemical composition of fraction f1-2, the molecules in this fraction were analyzed using an MS-based approach. Total ion chromatogram (TIC) profiles of fraction f1-2 were determined using a Q-Exactive Plus mass spectrometer (Thermo Scientific, San Jose, CA, USA) equipped with an UltiMate 3000 UHPLC system (Thermo Scientific, San Jose, CA, USA) in positive ion mode (Figure 4A, upper) and negative ion mode (Figure 4B, lower). The precise molecular weights of precursor ions and tandem mass product ions were used for molecule identification by Compound Discoverer software (v3.3; Thermo Scientific, San Jose, CA, USA) (Figure 4B). Apart from amino acids, nucleosides, and metabolites, small molecules such as catechin, adefovir, salsolinol, coclaurine, and its glucoside derivative were identified from fraction f1-2 (Figure 4C). To understand which compound is involved in EFHD2 inhibition, pure compounds of these molecules were used to treat H1299 cells. The results from Western blot indicated that coclaurine and argininosuccinic acid have EFHD2 inhibition activity in H1299 cells (Figure 4D). Argininosuccinic acid can be generated either from citrulline and aspartate by argininosuccinate synthetase or from fumarate and arginine by argininosuccinate lyase in the urea cycle [32]. Compared to argininosuccinate lyase, coclaurine is more commonly found in plant extract components; therefore, we prioritize coclaurine for exploring its roles in EFHD2 inhibition.

### 2.5. Characterization of Coclaurine in EFHD2 Inhibition and Cancer Biology

Our previous study indicated that EFHD2 activates the NOX4-ROS-ABCC1 pathway, thereby increasing resistance of NSCLC cells to cisplatin [11]. Accordingly, we investigated the effects of coclaurine on modulating this signaling pathway. Firstly, we assessed the cytotoxicity of coclaurine in NSCLC cells. The MTT assay revealed that coclaurine exhibits low toxicity in both H1299 and A549 cells, with IC50 values of 0.95 mM and 2 mM, respectively (Figure 5A). To minimize the impact of cytotoxic effects on the experimental results, we set the coclaurine dose at its IC90 value, which is 200 µM, for subsequent experimental analyses. As expected, coclaurine suppressed EFHD2 expression and attenuated the EFHD2-related signaling molecules ABCC1 and NOX4 as well as the intracellular ROS levels in NSCLC cells (Figure 5B,C). Additionally, coclaurine significantly sensitized NSCLC cells to cisplatin, reducing the IC50 value from 69.7 μM to 47.4 μM in H1299 cells and from 75.7 μM to 57.3 μM in A549 cells (Figure 5D). This result suggests that coclaurine could be a key molecule in *S. tetrandra* responsible for enhancing cisplatin sensitivity. Moreover, we examined the impact of coclaurine on NSCLC cancer biology, as its anticancer activities were largely unknown.

Coclaurine dose-dependently impaired the unlimited division of NSCLC cells in a colony formation assay [33] (Figure 5C). It also reduced the migration and invasion abilities of NSCLC cells in a dose-dependent manner (Figure 5D,E). Furthermore, coclaurine significantly decreased the spheroid formation in NSCLC cells (Figure 5F), suggesting that coclaurine could impair stemness properties, which are closely associated with chemoresistance and metastasis [34,35]. In line with this result, stemness-related markers CD44 [36] and epithelial cell adhesion molecule EpCAM [37] were reduced by coclaurine in Western blot analysis (Figure 5G). Collectively, our results suggest that coclaurine could not only sensitize NSCLC cells to cisplatin but also reduce the stemness and metastatic characteristics of NSCLC cells.

### 2.6. Coclaurine Suppresses EFHD2 Via Inhibiting Transcriptional Activity of FOXG1

To understand how coclaurine suppresses EFHD2, we conducted a pulse-chase experiment to determine the protein level alteration of EFHD2. NSCLC cells were treated with cycloheximide, an inhibitor of protein synthesis [38], either alone or in combination with coclaurine. Western blot analysis revealed that EFHD2 levels remained unchanged under both conditions (Figure 6A), suggesting that coclaurine does not affect the protein stability of EFHD2. Therefore, we investigated whether coclaurine impacts the gene expression of EFHD2. qPCR analyses showed that coclaurine significantly down-regulates EFHD2 mRNA levels (Figure 6B).

Next, we constructed sequential promoter sequences of EFHD2 into a vector with a luciferase reporter to verify the transcriptional activity (Figure 6C). Compared to the untreated control, the luciferase activity of the promoter region from −1001 to −1500 was most affected by coclaurine (Figure 6D). This region contains four transcription factor binding sites, including transcription factors Sp1 (SP1), forkhead box protein P1 (FOXP1) and G1 (FOXG1), and homeobox protein Hox-A13 (HOXA13) (Figure 6E). The ligand-binding domain plays a crucial role in the interaction between a kinase protein and its ligand. However, in drug development, most efforts targeting transcriptional regulators focus on the interface or pockets of the protein that interact with DNA [39]. To predict which transcription factor might interact with coclaurine, we performed a molecular docking analysis using BIOVIA Discovery Studio software (DS2022; RRID: SCR_015651). This analysis calculated multiple potential binding pockets for the ligand. The conformation with the most likely binding pocket and the lowest binding energy was selected as the final pose. The interaction between FOXG1 and coclaurine resulted in the greatest reduction in binding free energy (−36.90 kcal/mol) using the CDOCKER method (Figure 6E), suggesting that FOXG1 has a higher affinity for coclaurine than other transcription factors.

Molecular docking modeling revealed that coclaurine primarily binds to two pocket domains of FOXG1 to interfere with the interaction between FOXG1 and DNA: one domain containing amino acids K181, F184, and T270 (Figure 6F) and another domain containing N236, K237, C238, K272, and R274 (Figure 6G). To further confirm whether coclaurine impacts on FOXG1 protein stability, we examined the effect of coclaurine on the thermal stabilization of FOXG1 using a cellular thermal shift assay [40]. Western blot validation indicated that coclaurine decreases FOXG1 protein thermal stability from 70 °C to 60 °C (Figure 6H), suggesting that coclaurine may cause conformational changes in FOXG1, leading to its thermal instability.

To determine whether coclaurine inhibits EFHD2 via FOXG1, we knocked down FOXG1 (FOXG1/KD) using shRNA in H1299 cells. Western blot assay showed that FOXG1/KD partially suppresses EFHD2 compared to coclaurine treatment, suggesting that FOXG1 contributes to EFHD2 expression (Figure 6I). The combination of coclaurine and FOXG1/KD did not induce more EFHD2 inhibition compared to coclaurine alone (Figure 6I), indicating that coclaurine inhibits EFHD2 expression partially by interrupting FOXG1 activity. Functional assays revealed that FOXG1/KD sensitizes NSCLC cells to cisplatin, while the combination of coclaurine and FOXG1/KD does not increase the sensitivity of NSCLC cells to cisplatin compared to coclaurine treatment alone (Figure 6J), consistent with the Western blot finding. In summary, coclaurine suppresses EFHD2 expression partially by modulating the transcriptional activity of FOXG1.

## 3. Discussion

Due to advanced screening techniques and the increasing use of regular screening, the prevalence of NSCLC diagnosis at an early stage has increased [41]. Surgical resection is the primary curative treatment for these patients [3]. However, approximately one-third to one-half of patients develop recurrent disease without additional treatment [42]. Consequently, adjuvant strategies have been considered for patients who have high-risk clinical features of recurrence, such as undifferentiated status, large tumor size, and vascular invasion [5]. Molecular markers have also been investigated to predict the risk of recurrence, including circulating DNA and RNA [43,44], cell cycle and immune-related genes [45,46], and alteration of specific protein expression [10,47]. Targeted therapy, such as inhibitors of mutant EGFR, serves as an available adjuvant approach, but only a small proportion of patients have targetable mutations [48]. Immunotherapy with immune-checkpoint inhibitor (ICI) has been exploited therapeutically to maximize clinical outcomes in patients with advanced NSCLC [49]. Nevertheless, determining the optimal combination, sequence, and duration of ICI treatment and evaluating robust biomarkers and endpoints in the adjuvant settings remains challenging [50]. Cisplatin-based adjuvant therapy is currently the standard of care for completely resected NSCLC. Unfortunately, such treatment only improves patient outcome with approximately a 4% increase in 5-year survival [6]. Numerous factors may contribute to this unsatisfactory outcome, including tumor intrinsic and acquired chemoresistance, lack of biomarker for treatment responsiveness, and drug toxicity. The earlier analyses indicated that approximately 50% of these patients do not complete the entire adjuvant treatment due to toxicity, including neutropenia, fatigue, dyspnea, pain, and loss of appetite [51,52]. Therefore, an increase in sensitivity is critical for lowering drug dosage and improving responsiveness, thereby enhancing the effectiveness of adjuvant chemotherapy. Our previous studies demonstrated that the expression of EFHD2 is not only significantly associated with postsurgical recurrence in patients with stage I NSCLC [10], but also increases NSCLC resistance to cisplatin through activating the NOX4-ROS-ABCC1 signaling pathway [11]. Accordingly, EFHD2-targeting strategies may provide an alternate approach for enhancing patient responsiveness to cisplatin-based adjuvant chemotherapy.

Combining chemotherapeutic drugs to reduce resistance and synergically increase efficacy is broadly applied in various cancers. However, the accumulation of toxicity from each drug, especially for long-term treatment, can cause deleterious systemic effects [19]. As a result, natural-based alternatives with lower toxicity have emerged as an attractive regimen for adjuvant chemotherapy. Herbal medicine is a crucial component of Traditional Chinese Medicine (TCM), which typically includes multiple herbs and ingredients necessary for efficacy [53]. Recently, Chinese medicine prescriptions such as Shenyi capsule and Yifei Qinghua granules have been clinically evaluated for adjuvant efficacy, showing improvements in survival outcomes [54]. Unlike double-blind tests, the grouping of recruited subjects at postoperative stage of NSCLC and Chinese medicine intervention were based on the main syndromic types and constitutions of individual patients, categorized into pulmonary qi deficiency, qi and yin deficiency, and stagnation of phlegm and blood stasis according to the eight principles in TCM theory [55,56]. Traditionally, the primary purpose of natural compounds as chemotherapeutic sensitizers is to broaden the therapeutic window of chemotherapeutic drugs and reduce the incidence of chemotherapy resistance. The functional mechanisms of these compounds mainly involve targeting drug-induced oxidative stress and NF-κB signaling to increase cytotoxicity [57]. Apart from mixtures, several single herbal molecules were reported to sensitize NSCLC cells to chemotherapy, including celastrol [58], curcumin [59], ethoxysanguinarine [60], osthole [61], phloretin [62], and resveratrol [63]. Based on evidence of their molecular functions, an alternative EFHD2-targeting strategy was used to screen potential compounds from herbal extracts. In this study, we found that coclaurine, a compound in *S. tetrandra*, effectively sensitizes NSCLC cells to cisplatin and also inhibits tumorigenesis and stemness characteristics in NSCLC cells, suggesting that coclaurine may serve as a lead compound for developing EFHD2 inhibitors.

*S. tetrandra*, a medicinal herb widely used in traditional Chinese medicine, has garnered increasing attention for its anticancer properties. Its bioactive compounds, particularly bisbenzylisoquinoline alkaloids, exhibit promising potential in cancer treatment [64]. Tetrandrine induces cell cycle G1 arrest via increasing the degradation of G1-S-specific cyclin-dependent kinases and apoptosis via activation of caspase and cleavage of poly (ADP ribose) polymerase in various cancer cells, including colon cancer, liver cancer, oral cancer, and pancreatic cancer [27,65,66,67]. Another main bioactive compound, fangchinoline, and its derivatives can induce cell cycle G1 arrest via suppression of the PI3K/AKT and MAPK signaling pathways in vitro and in vivo [68,69]. *S. tetrandra* has been demonstrated to reverse cisplatin resistance in lung cancer xenografts through downregulating multidrug resistance-associated proteins [23]. Both tetrandrine and fangchinoline also enhance the efficacy of cisplatin in the treatment of lung cancer [70,71]. Moreover, tetrandrine/CBT-01^®^ and its 5-bromo derivative has been applied to the stage of clinical trials [72]. These studies suggest the potential of *S. tetrandra* and its active components as a chemosensitizer to improve treatment.

Coclaurine is one of the benzylisoquinoline alkaloids, which are recognized as sources of pharmacologically significant chemical components in both organic and medicinal chemistry [73]. Despite their structural diversity, quinoline alkaloid derivatives obtained from different plants have been reported to exhibit antineoplastic effects on various tumor cells [74,75]. Coclaurine, identified from the sugar apple plant *Annona squamosal*, induces apoptosis in breast cancer cells by increasing the expression of p53, BAX, and caspase 3 and 9 proteins [76]. The bioavailability of coclaurine has not been extensively studied in detail. A pharmacokinetic study demonstrated that when aqueous extracts of Ziziphi Spinosae Semen containing coclaurine were administered intragastrically at a dose of 6.8 g/kg in a Sprague Dawley rat model, the Cmax of coclaurine was reached at 0.3 h post-administration in the plasma, suggesting rapid absorption from the gastrointestinal tract [77]. Additionally, coclaurine was rapidly eliminated from the rat plasma after intragastric administration, with a T1/2 of 0.45 ± 0.17 h, indicating a short duration of action [77]. According to the general principles of alkaloid metabolism, the liver and gut likely play significant roles in phase I and II metabolic transformations of coclaurine, mediated by cytochrome P450 enzymes and uridine diphosphate glycosyltransferases [78]. Due to its rapid absorption and elimination, achieving a 200 μM concentration of coclaurine in human plasma is unlikely. Thus, developing more effective derivatives of coclaurine may be a more practical approach for clinical application.

Several mechanisms are involved in the development of cisplatin resistance, such as decreased intracellular drug accumulation and decreased mismatch-repair activity. Accumulating data reveal that transcription factors potentially contribute to generating resistance in various cancers [79]. Despite lack of the HMG domain, which is commonly associated with proteins involved in recognizing DNA lesions, both transcriptional factors Y-box binding protein 1 (YB-1) and Zinc Finger Protein 143 (ZNF143) can bind to cisplatin-modified DNA, indicating their potential roles in recognizing DNA damage and initiating repair processes [80]. Transcriptional factor Homeobox D8 (HOXD8) is significantly overexpressed in patients with recurrent ovarian cancer when compared to patients with primary malignant tumors and is associated with cisplatin resistance and metastasis in the advanced disease [81]. Additionally, epigenetic activation of FOXF1 confers cancer stem cell properties promotes cisplatin resistance in NSCLC [82]. A recent study showed that a histone deacetylase inhibitor, trichostatin A, which induces SNAI1 and SNAI2 expression, inhibits SLC2A5/GLUT5 expression and thereby sensitizes colon cancer cells to cisplatin and oxaliplatin [83]. Here, we found a novel function of coclaurine in regulating the stability of transcription factor FOXG1, thereby inhibiting EFHD2 expression (Figure 6). FOXG1, a member of the winged-helix forkhead family, is mainly expressed in brain tissue and acts as a transcriptional repressor in the development of the telencephalon [84]. A recent report indicated that FOXG1 drives transcriptomic networks, including suppressing *Zbtb20*, *Prox1*, and *Epha4* and activating *Nr4a2* during the development of the medial pallium [85], suggesting that FOXG1 can serve as a transcriptional promoter or repressor depending on the gene promoter sequence and structure. Emerging evidence revealed that FOXG1 is overexpressed in numerous types of cancers, such as glioblastoma, breast cancer, liver cancer, nasopharyngeal cancer, and lung cancer [86,87,88,89,90]. It is involved in the negative regulation of cell apoptosis and promotes tumor proliferation, likely through the PI3K-AKT pathway [90,91]. However, the role of FOXG1 in chemoresistance remains uncharacterized. In this study, we uncovered the novel function of FOXG1 in activating EFHD2 suppression, thereby increasing NSCLC cell resistance to cisplatin treatment.

## 4. Materials and Methods

### 4.1. Cell Culture

Human lung epithelial carcinoma cell lines A549 and H1299, as well as the human non-tumorigenic bronchial epithelial cell line BEAS-2B, were purchased from the American Type Culture Collection (ATCC, Manassas, VA, USA). A549 cells were maintained in RPMI 1640 medium (#11875127; Gibco, Waltham, MA, USA), while H1299 cells were cultured in DMEM/F-12 medium (#11320082; Gibco, Waltham, MA, USA). Both media were supplemented with 10% fetal bovine serum (FBS; #26140079; Gibco, Waltham, MA, USA) and 1% penicillin-streptomycin (#15140122; Gibco, Waltham, MA, USA). BEAS-2B cells were cultured in DMEM/F-12 medium, supplemented with 1 μM hydrocortisone, 5 μg/mL insulin, 10 μM HEPES, 10% fetal bovine serum, and 1% penicillin-streptomycin. All cell lines were grown in a humidified atmosphere of 5% CO_2_ and 95% air at 37 °C.

### 4.2. Aqueous Extraction for the Herbal Materials

Twelve types of herbal materials were purchased from local Chinese medicine shops (Table 1). Prior to aqueous extraction, the herbal raw materials were cut into small pieces (length < 1 cm) to enhance extraction efficiency. Fragmented herbal materials (2 g) were incubated in 25 mL of double distilled water with stirring at 170 °C for 30 min. High-speed centrifugation (10,000× *g*) was used to remove precipitates and residues, and the suspensions were filtered through a 0.22 μm pore size membrane (#6534; Sartorius, Göttingen, Germany). After speed-vacuum drying, the contents of the aqueous extracts were determined by weighting. The filtered products were stored in aliquots at −20 °C.

### 4.3. Cell Viability Assay

Cell viability was determined using the methylthiazol tetrazolium (MTT) method. After treatment with natural compounds or pharmaceutical drugs, 3-(4,5-dimethylthiazol-2-yl)-2,5-diphenyltetrazolium bromide (#M6494; Invitrogen, Carlsbad, CA, USA) was added to the cultured cells, and the reduced formazan compounds were measured at a wavelength 570 nm with an enzyme-linked immunosorbent assay reader (Dynex, Chantilly, VA, USA). Cell viability (%) was calculated as (T/U) × 100%, where T is the absorbance of treated cells and U is the absorbance of untreated cells.

### 4.4. Western Blot Assay

Total proteins from culture cells were extracted using the RIPA lysis and Extraction buffer (#89901; Thermo Scientific, Waltham, MA, USA) supplemented with a proteinase inhibitor cocktail (Sigma-Aldrich, St. Louis, MO, USA). Protein concentration was determined using the Protein Assay kit (#5000006EDU; Bio-Rad, Hercules, CA, USA) at 595 nm absorbance. A total of 20 μg of protein was separated by 9.5% SDS-PAGE and then transferred onto PVDF membranes (#88518; Thermo Scientific, Waltham, MA, USA) at 400 mA for 3 h. After blocking with a 5% skin milk solution, the membranes were incubated with primary antibodies at 4 °C for 16–24 h. The primary antibodies used in this study are summarized as follows: EFHD2 (ab106667), NOX4 (ab133303), and CD44 (ab51037) purchased from Abcam (Cambridge, UK); ABCC1 (#72202), ALDH1A1 (#12035), EpCAM (#93790), FOXG1 (#29642), and β-actin (#4970) purchased from Cell Signaling (Danvers, MA, USA). The membranes were washed with TBST buffer and agitated gently three times for 15 min each. The reaction with horseradish peroxidase-conjugated secondary antibodies was performed at room temperature for 1 h. Immunoreactive signals were revealed using an enhanced ECL substrate Western Lighting Plus-ECL (PerkinElmer, Shelton, CT, USA) and recorded by developing photographic film under optimal exposure or with the luminescence image analyzer ImageQuant LAS 4000 (GE Healthcare Life Sciences, Washington, DC, USA). The original images of the Western blot assays are shown in Appendix A.

### 4.5. Transwell Migration and Invasion Assays

For in vitro migration assays, tumor cells (3 × 10^4^ cells in 200 μL) were suspended in the upper half of a PET membrane transwell insert chamber (#353097; Corning, Corning, NY, USA) on a 24-well plate. For in vitro invasion assays, tumor cells (3 × 10^4^ cells in 200 μL) were suspended in transwell insert chambers coated with matrigel (1 mg/mL; #356234; BD Biosciences, Franklin Lakes, NJ, USA). Media without FBS supplementation were added to the upper chamber, while media with 10% FBS supplementation were added to the lower chamber. After incubation at 37 °C for 16 h or 24 h for migration and invasion assays, respectively, tumor cells that had passed through the insert were fixed with 3.7% formalin (#15512; Sigma-Aldrich, St. Louis, MO, USA) and stained with 0.1% crystal violet (#C0775; Sigma-Aldrich, St. Louis, MO, USA). For quantification, crystal violet was extracted using 50% ethanol (#459836; Sigma-Aldrich, St. Louis, MO, USA) and 0.1% acetic acid (#A6283; Sigma-Aldrich, St. Louis, MO, USA) and subjected to colorimetric measurement at 570 nm.

### 4.6. Colony Formation Assay

For the colony formation assay, tumor cells (1000 cells per well) were seeded into 6-well plates and then left treated or untreated (control) with the indicated doses of coclaurine. After 10 days of culture, the cells were fixed with 3.7% formalin and stained with 0.1% crystal violet. For quantification, the stained dye was extracted using 50% ethanol and 0.1% acetic acid and measured by colorimetry at 570 nm.

### 4.7. Spheroid Formation Assay

Tumor cells (2000 cells per dish), either treated or untreated (control) with the indicated doses of coclaurine, were cultured in 6 cm culture dishes coated with 1% agarose for 7 d. The number of spheroid formations was counted manually under a bright-field microscope.

### 4.8. MS Identification of Small Molecules

An Ultimate 3000 UHPLC system (Thermo Scientific, San Jose, CA, USA) coupled with a Q-Exactive Plus high-resolution mass spectrometer (Thermo Scientific, San Jose, CA, USA), equipped with a standard heated electrospray ionization source (HESI), was used for molecule identification. The chromatographic conditions for the separation of herbal materials (10 μL) were set as follows: flow rate at 300 μL/min, with 0.1% formic acid (Fluka 56302; Honeywell, Charlotte, NC, USA) in water as mobile phase A and acetonitrile (UN1648; J.T.Baker, Radnor, PA, USA) as mobile phase B, using an ACQUITY UPLC C18 column (particle size 1.7 μm, 2.1 mm × 100 mm; Waters, Milford, MA, USA). The HESI voltage was maintained at 4.0 kV for positive ionization mode and 3.5 kV for negative ionization mode. A full scan-top N data-dependent acquisition (full MS/ddMS^2^) scan mode was used to detect the components in the herbal materials. The mass scan range was *m*/*z* 150–1500 with a mass resolution of 70,000, and tandem mass (MS^2^) resolution was 17,500. Full scan spectra and product ion spectra were utilized to identify herbal phytochemicals using Compound Discoverer v3.3 software (Thermo Scientific, San Jose, CA, USA), which integrates with the databases ChemSpider, PlantCyc, mzVault, and mzCloud.

### 4.9. Protein Thermal Shift Assay

The effect of coclaurine on the thermal stabilization of the target transcriptional factor was determined by a cellular thermal shift assay [40]. Briefly, tumor cells (5 × 10^6^ cells in 1 mL) were treated or left untreated with coclaurine (200 μM) at 37 °C for 2 h. After washing, the tumor cells were suspended in 400 μL of PBS supplemented with protease inhibitors. The tumor cells were then aliquoted into PCR tubes (50 μL per vial) and heated at different temperatures (30 °C, 40 °C, 50 °C, 60 °C, and 70 °C) in a PCR machine for 3 min. After cooling at room temperature for 3 min, the tumor cells were immediately frozen with liquid nitrogen. Once all reactions were completed, the tumor cells were mixed with SDS-PAGE sample buffer and heated at 100 °C for 10 min. The samples were centrifuged at 12,000 rpm at 4 °C for 10 min, and the supernatant (10 μL) of each sample was performed using electrophoresis under reducing condition and Western blot validation.

### 4.10. Cloning of the EFHD2 Gene Promoter

For the EFHD2 gene promoter assay, primers were designed to target the corresponding sequences, with a KpnI restriction enzyme site added to the forward primer and a HindIII restriction enzyme site added to the reverse primer, along with a three-nucleotide overhang at the 5′ end of each primer. The information of primer sequences was listed in Table 2. Target sequences were amplified by the paired primers and constructed into the pGL4 firefly luciferase vector using the dual-luciferase reporter assay system (Promega). Luciferase activities were determined by a luminometer using a dual-luciferase reporter assay according to the instructions of the manufacturer.

### 4.11. Molecular Docking

Molecular docking was performed to estimate the binding affinity of coclaurine to the target transcriptional factors using BIOVIA Discovery Studio software (DS2022; RRID: SCR_015651). The three-dimensional structure of coclaurine was obtained from PubChem (accessed on 3 January 2024; https://pubchem.ncbi.nlm.nih.gov), and the molecular formula of coclaurine was transformed into canonical SMILES as follows: COC1=C(C=C2C(NCCC2=C1)CC3=CC=C(C=C3)O)O [92]. The protein structures of transcriptional factors FOXG1 (7CBY), FOXP1 (2KIU), HOXA13 (2L7Z), and SP1 (1SP1) were acquired from the Protein Data Bank (accessed on 4 January 2024; https://www.rcsb.org/) and used for molecular docking calculations by the CDOCK method. The resultant structures from molecular docking were output and presented using DS2022 and the PyMOL Molecular Graphics System (RRID: SCR_000305).

### 4.12. Statistics

The quantitative characteristics of the data were displayed as mean and standard deviation (SD). The statistical significance of the difference between the responses of two groups was analyzed using a two-tailed Student’s *t*-test, which was applied for the MTT assay and qPCR assay. One-way ANOVA followed by Tukey’s post hoc test was used for the comparison of multiple groups, such as cell survival, cell migration, invasion, spheroid formation and colony formation assays. Statistical analyses were performed using IBM SPSS Statistics 22. A *p*-value of <0.05 was considered statistically significant.

## 5. Conclusions

We demonstrate that the aqueous extracts of *Stephania tetrandra* effectively inhibit EFHD2 expression in NSCLC cells, highlighting its potential as a therapeutic agent. Coclaurine, one of the key active compounds identified in *S. tetrandra*, plays a crucial role in this inhibition. Furthermore, coclaurine disrupts EFHD2-related NOX4-ABCC1 signaling, enhancing the sensitivity of NSCLC cells to cisplatin treatment. Mechanistically, coclaurine interferes with the binding of FOXG1 to the EFHD2 promoter, leading to a significant reduction in EFHD2 mRNA transcription. The representative working model is shown in Figure 7. These findings strongly suggest that *S. tetrandra* and coclaurine may serve as promising adjuvant therapies to enhance the efficacy of cisplatin in NSCLC treatment.

## Figures and Tables

**Figure 1 pharmaceuticals-17-01356-f001:**
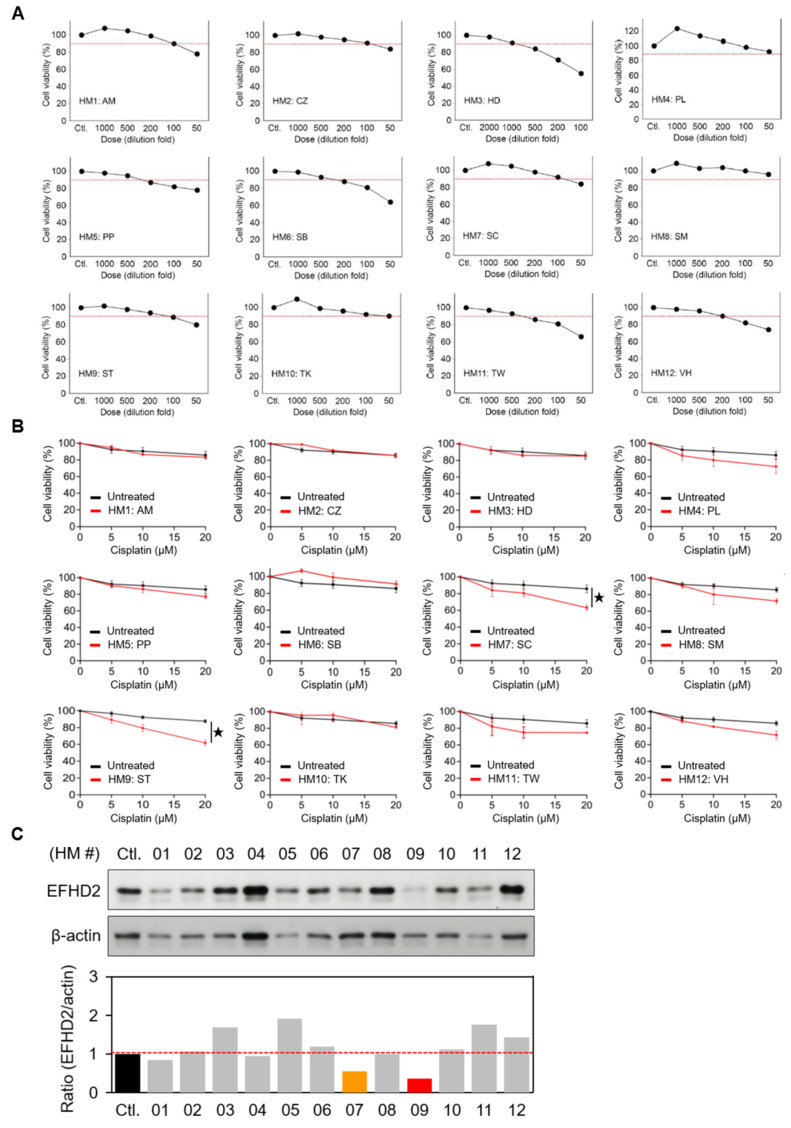
Aqueous extracts of *S. tetrandra* inhibit EFHD2 expression. (**A**) Cell viabilities of H1299 cells treated with the indicated dilution doses of aqueous herbal extracts were determined by MTT assay. The red dash line represents 90% of cell viability (*n* = 3). (**B**) Cell viabilities of H1299 cells treated with or without the IC90 dose of each herbal extract, followed by the indicated doses of cisplatin, were determined by MTT assay (*n* = 3). (**C**) EFHD2 expression in H1299 cells treated with the IC90 dose of each herbal for 24 h was determined by Western blot. β-actin, loading control. The relative expression levels are shown as the ratio of EFHD2 to β-actin, with the untreated control group serving as the normalized control. Data are displayed as the means  ±  SD. For statistical analysis, a 2-tailed unpaired Student’s *t*-test was used (**B**). ^★^, *p* < 0.05.

**Figure 2 pharmaceuticals-17-01356-f002:**
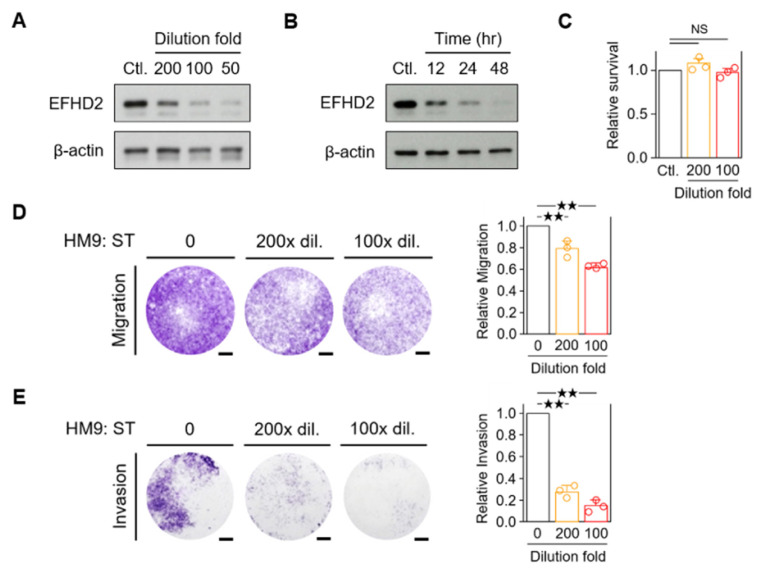
Aqueous extracts of *S. tetrandra* suppress the migration and invasion of NSCLC cells. (**A**) EFHD2 levels in H1299 cells treated with the indicated dilution doses of *S. tetrandra* were determined by Western blot. (**B**) EFHD2 levels in H1299 cells treated with a 100× dilution dose of *S. tetrandra* for the indicated times were determined by Western blot. β-actin, loading control. (**C**) Cell viability of BEAS-2B cells treated with the indicated dilution doses of *S. tetrandra* was determined by MTT assay. Relative cell survival was normalized to the untreated control (*n* = 3). (**D**,**E**) Cell migration and invasion assays of H1299 cells were conducted with the indicated dilution doses of *S. tetrandra*. Signal quantification with crystal violet extract was measured by colorimetric analysis at 570 nm. Relative migration/invasion ability was normalized to the untreated control (*n* = 3). Scale bar, 0.1 cm. Data are displayed as the means  ±  SD. NS, no significant. For statistical analysis, one-way ANOVA with Tukey’s post hoc test (**D**,**E**). ^★★^, *p* < 0.01.

**Figure 3 pharmaceuticals-17-01356-f003:**
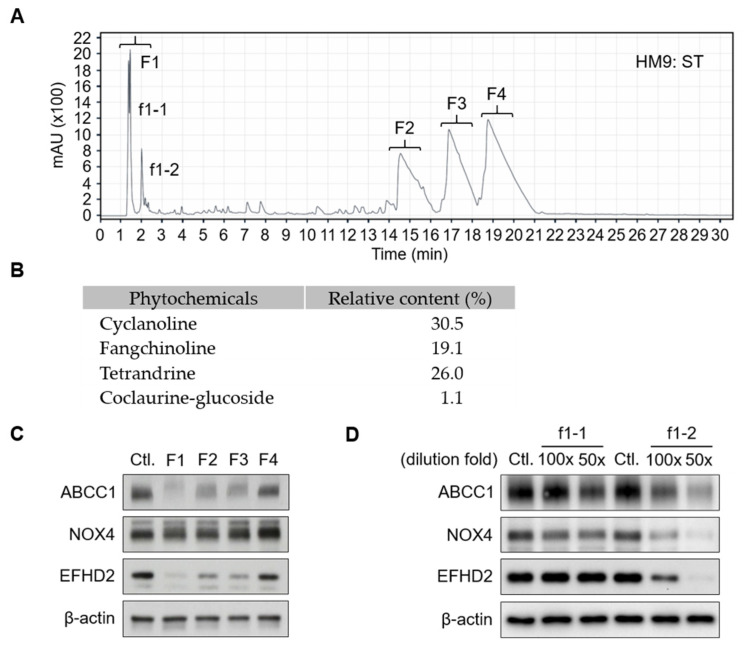
HPLC purification of aqueous extracts of S. tetrandra. (**A**) Aqueous extracts of S. tetrandra (50 μL) were subjected to C18 HPLC separation using an acetonitrile gradient: 5–30% (0–30 min); 30–60% (30–50 min); 60–5% (50–60 min). The effluent was monitored at 214 nm. The eluate was dried using a vacuum centrifugation concentrator and reconstructed with the same volume of water (50 μL). (**B**) The relative content of the major peaks (F2–F4) along with coclaurine-glucoside was calculated by the ratio of peak area of each phytochemical to the total peak areas in the chromatogram monitored at 280 nm. (**C**) A 100× dilution of fractions F1–F4 were used to treat H1299 cells for 24 h. (**D**) Fractions f1-2 and f1-2 were further purified and used to treat H1299 cells at the indicated dilution doses for 24 h. The protein expression levels of ABCC1, NOX4, and EFHD2 were determined by Western blot. Untreated H1299 cells served as the protein expression control. β-actin, loading control.

**Figure 4 pharmaceuticals-17-01356-f004:**
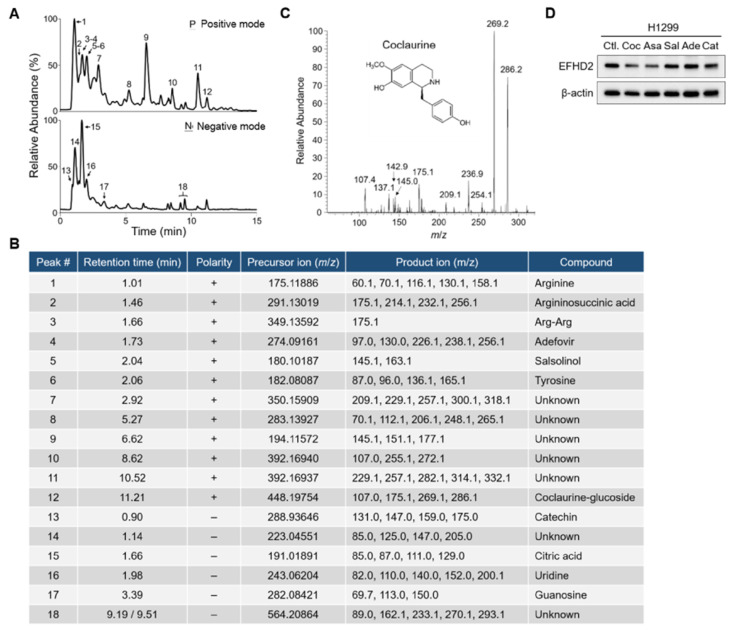
Coclaurine is a critical compound of fraction f1-2 responsible for EFHD2 inhibition. (**A**) Fraction f1-2 was analyzed by an UHPLC-Q-Exactive Plus MS using both positive ion mode (upper) and negative ion mode (lower). (**B)** The identified compounds corresponding to the indicated peaks of the TIC profiles in (**A**). (**C**) The tandem mass profile and molecular structure of coclaurine. (**D**) EFHD2 expression in H1299 cells treated with pure compounds identified from fraction f1-2 was determined by Western blot. β-actin, loading control. Coc, coclaurine; Asa, argininosuccinic acid; Sal, salsolinol; Ade, adefovir; Cat, catechin.

**Figure 5 pharmaceuticals-17-01356-f005:**
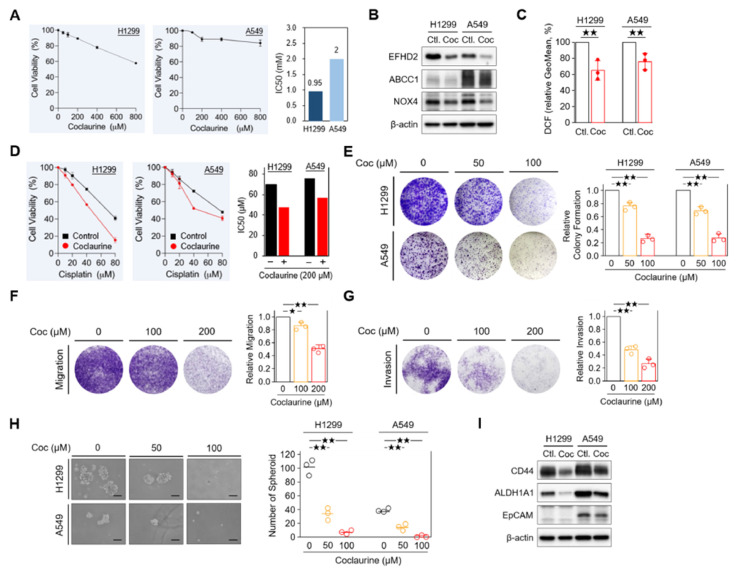
Characterization of coclaurine in EFHD2 inhibition and cancer biology. (**A**) Cell viability of H1299 and A549 cells treated the indicated doses of coclaurine was analyzed by MTT assay. (**B**) The expression of EFHD2 and its related signaling molecules in H1299 and A549 cells treated or untreated with coclaurine (200 μM) were determined by Western blot. (**C**) The intracellular ROS levels in H1299 and A549 cells treated or untreated with coclaurine were determined using the tracer dye CM-H2DCFDA (DCF, Invitrogen). (**D**) Cell viability of H1299 and A549 cells treated or untreated with coclaurine (200 μM) at the indicated doses of cisplatin was analyzed by MTT assay. (**E**) Colony formation assay in H1299 and A549 cells and (**F**,**G**) cell migration assay and invasion assay in H1299 cells treated with the indicated doses of coclaurine. Signal quantification using crystal violet extract was measured by colorimetric analysis at 570 nm. The relative signal intensities were normalized to the untreated control (*n* = 3). Scale bar, 0.1 cm. (**H**) H1299 and A549 cells were treated with the indicated doses of coclaurine for 10 d, the number of spheroid formation was counted manually under a bright-field microscope (*n* = 3). Scale bar, 100 μm. (**I**) The expression of stemness-related proteins in H1299 and A549 cells treated or untreated with coclaurine (200 μM) was determined by Western blot. β-actin, loading control. Data are displayed as the means  ±  SD. For statistical analysis, a 2-tailed unpaired Student’s *t*-test (**C**,**E**–**H**). ^★^, *p* < 0.05; ^★★^, *p* < 0.01.

**Figure 6 pharmaceuticals-17-01356-f006:**
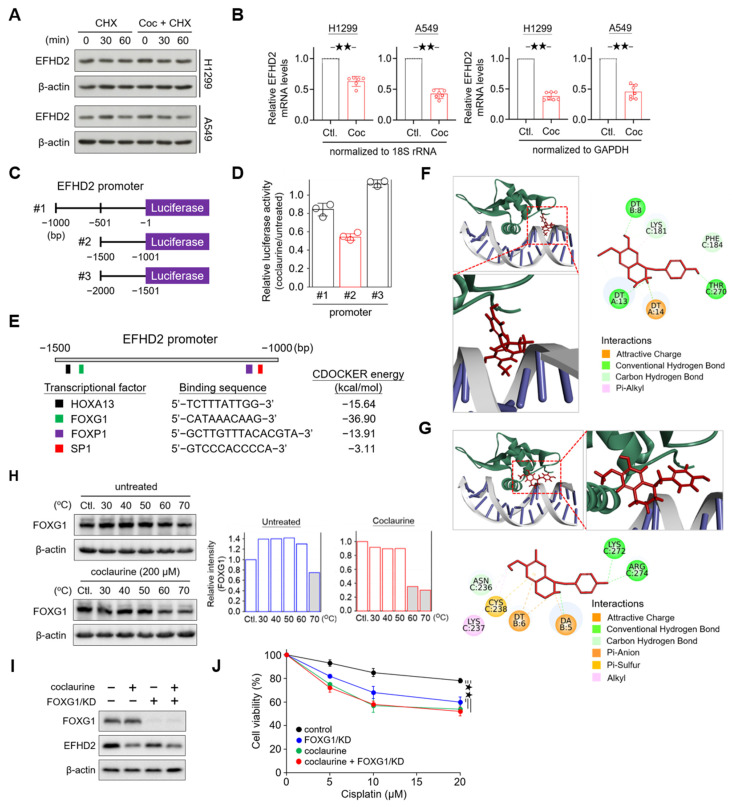
Coclaurine suppresses EFHD2 through inhibition of the transcriptional activity of FOXG1. (**A**) After cycloheximide (CHX, 10 mg/mL) treatment for 1 h, H1299 and A549 cells were treated or untreated with coclaurine (200 μM) for the indicated times. EFHD2 expression in each experimental group was determined by Western blot. (**B**) EFHD2 mRNA levels of H1299 and A549 cells treated or untreated with coclaurine (200 μM) for 24 h were determined by qPCR. GADHP and 18S rRNA served as internal controls for gene expression. (**C**) Schematic illustration showing the construction of the EFHD2 promoter sequence in the luciferase expression vector. (**D**) Relative luciferase activity with various EFHD2 promoter sequences in H1299 cells treated with coclaurine (200 μM) compared to the untreated control. (**E**) Schematic illustration showing the binding sites of transcription factors and the alteration of CDOCKER energy by coclaurine binding. (**F**,**G**) Three- and two-dimensional interactions between transcription factor FOXG1 (green), DNA (gray), and coclaurine (red) in two potential binding pockets. The binding amino acids in FOXG1 and DNA nucleotides with coclaurine are shown in the two-dimensional structures. (**H**) FOXG1 thermal stability in H1299 cells treated or untreated with coclaurine (200 μM) at the indicated temperatures was determined by Western blot. The relative FOXG1 level (FOXG1/actin) was normalized to the unheated control of each group. (**I**) FOXG1 and EFHD2 expression and (**J**) cell viability of H1299 cells with or without FOXG1/KD in the presence or absence of coclaurine (200 μM) were determined by Western blot and MTT assay, respectively. Data are displayed as the means  ±  SD. For statistical analysis, a 2-tailed unpaired Student’s *t*-test (**J**). ^★★^, *p* < 0.01.

**Figure 7 pharmaceuticals-17-01356-f007:**
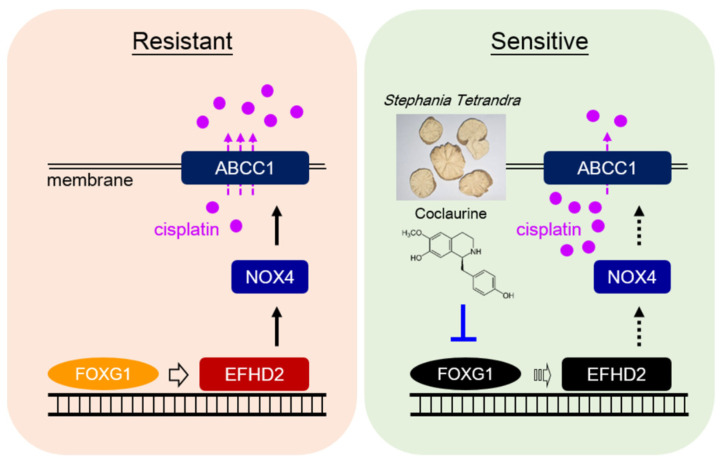
Representative working model. The aqueous extracts of *S. tetrandra* and its effective compound coclaurine can interfere with the interaction between transcription factor FOXG1 and the EFHD2 promoter, thereby reducing EFHD2 mRNA levels and its transcription. Consequently, these treatments attenuate the expression of EFHD2-related signaling molecules NOX4 and ABCC1, leading to decrease cisplatin efflux and ultimately sensitize NSCLC cells to cisplatin.

**Table 1 pharmaceuticals-17-01356-t001:** The contents in the aqueous extractions of Chinese herbs.

NO.	Herbal Medicine	Symbol	Chinese Name	Conc. (mg/mL)	IC90 (Dilution Fold)
HM 01	*Atractylodes macrocephala*	AM	白朮	57	100×
HM 02	*Curcuma zedoaria*	CZ	莪術	12	200×
HM 03	*Hedyotis diffusa*	HD	白花蛇舌草	29	3000×
HM 04	*Paeonia lactiflora*	PL	赤芍	44	100×
HM 05	*Paris polyphylla*	PP	蚤休	28	500×
HM 06	*Scutellaria barbata D. Don*	SB	半枝蓮	27	200×
HM 07	*Smilax china*	SC	菝葜	19	100×
HM 08	*Salvia miltiorrhiza*	SM	丹參	26	100×
HM 09	*Stephania tetrandra S.Moore*	ST	石蟾蜍	20	200×
HM 10	*Trichosanthes kirilowii*	TK	栝蔞	33	100×
HM 11	*Tripterygium wilfordii*	TW	雷公藤	18	500×
HM 12	*Vaccaria hispanica*	VH	王不留行	9	500×

**Table 2 pharmaceuticals-17-01356-t002:** The information of primer sequences.

Primer	Sequence	%GC	Tm (°C)
#1-F	GCGCGGTACCAAAATCTGTATTATTAACCTAGGCCCCAGT	37	67
#1-R	GCGCAAGCTTGGTGGCCCGCGCGGCACTCGCCTTGGCCGG	83	88
#2-F	GCGCGGTACCTTTAAAACAAAAAAACCATTTTAGCTGAGC	27	64
#2-R	GCGCAAGCTTCAGGCTCCAATTTGGACAGACGAA	50	68
#3-F	GCGCGGTACCACAAAGCCAAATAGGTAACTGACCAGAGAG	43	69
#3-R	GCGCAAGCTTTCTCAGAATTAAGTTTCTCTTAGTTTTGGG	33	65

## Data Availability

Data are contained within the article and Appendix A.

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
