# Peer review of "Stephania tetrandra and Its Active Compound Coclaurine Sensitize NSCLC Cells to Cisplatin through EFHD2 Inhibition"

_pharmaceuticals, 2024, doi:10.3390/ph17101356_

Round 1
Reviewer 1 Report
Comments and Suggestions for Authors
1. The abstract should be revised to include the major findings of the study, ensuring it is written in a scientific manner.
2. Given the relevance of the study, the following articles contain important information about EFHD2 and its role in SCLC; therefore, the authors should include these references in the manuscript.
Fan CC, Tsai ST, Lin CY, Chang LC, Yang JC, Chen GY, Sher YP, Wang SC, Hsiao M, Chang WC. EFHD2 contributes to non-small cell lung cancer cisplatin resistance by the activation of NOX4-ROS-ABCC1 axis. Redox Biol. 2020 Jul;34:101571.
Wu J, Xu X, Duan J, Chai Y, Song J, Gong D, Wang B, Hu Y, Han T, Ding Y, Liu Y, Li J, Cao X. EFHD2 suppresses intestinal inflammation by blocking intestinal epithelial cell TNFR1 internalization and cell death. Nat Commun. 2024 Feb 12;15(1):1282.
3. In the introduction section, discuss the chemical properties and QSAR activity of Stephania tetrandra S. Moore (S. tetrandra) that contribute to its therapeutic effects against various diseases.
4. The authors need to quantify each ingredient present in the extract of Stephania tetrandra S. Moore (S. tetrandra) in this study.
5. The authors need to provide the original experimental data represented in Fig. 1 and Fig. 2. The data must be analyzed using appropriate statistical tools.
6. The representation of the results lacks statistical modeling, which undermines the validity of the work. The authors need to validate the results using appropriate statistical models for each study mentioned in the manuscript.
7. The discussion section requires extensive revisions. The authors should correlate the findings with existing literature and highlight the novelty of the investigation.
8. The conclusion needs to be revised to emphasize the major findings.

Comments on the Quality of English LanguageIt's fine.
Reviewer 2 Report
Comments and Suggestions for Authors
This is a very comprehensive study by Shu-Yu Hu et al. to observe the potential of S. tetrandra as an adjuvant therapy for cisplatin-resistant NSCLC patients. The authors have successfully shown that coclaurine, a key compound in S. tetrandra, suppressed EFHD2 expression by disrupting the FOXG1-EFHD2 transcriptional interaction. Coclaurine not only enhanced cisplatin sensitivity but also reduced stemness and metastasis in NSCLC cells. Their findings suggest that S. tetrandra and coclaurine could improve cisplatin efficacy in NSCLC treatment. This manuscript can be accepted for publication.
1. What is the main question addressed by the research?
The potential of S. tetrandra as an adjuvant therapy for cisplatin-resistant NSCLC patients.
2. Do you consider the topic original or relevant in the field? Does it address a specific gap in the field? Please also explain why this is/ is not the case.
Yes, the topic is relevant for the field. The authors have successfully shown that coclaurine, a key compound in S. tetrandra, suppressed EFHD2 expression by disrupting the FOXG1-EFHD2 transcriptional interaction. Coclaurine not only enhanced cisplatin sensitivity but also reduced stemness and metastasis in NSCLC cells. This is a novel finding that is highly relevant to the field of study.
3. What does it add to the subject area compared with other published material?
Their findings suggest that S. tetrandra and coclaurine could improve cisplatin efficacy in NSCLC treatment. The study addresses the involvement of oxidative stress in the stem/precursor thyroid cells that inhibits differentiation.
4. What specific improvements should the authors consider regarding the methodology? What further controls should be considered?
None as such
5. Are the conclusions consistent with the evidence and arguments presented and do they address the main question posed? Please also explain why this is/is not the case. Yes, the authors have used appropriate methods to evaluate their research questions and the findings corroborate the same.
6. Are the references appropriate? Yes.
7. Please include any additional comments on the tables and figures. The figures and tables provided in the manuscript and how they are organized are among the best representations I have seen this year.
Reviewer 3 Report
Comments and Suggestions for Authors
This study by Hu et al investigated about the herbs, Stephania tetrandra (S. tetrandra) exhibited the best efficacy in EFHD2 inhibition correlation with cisplatin sensitization. Coclaurine was identified as the most critical compound for EFHD2 inhibition through LC/MS and some assays. It enhanced cisplatin-induced cell death by EFHD2-related NOX4-ABCC1 signaling. Coclaurine interfered with the interaction between transcriptional factor FOXG1 and the EFHD2 promoter sequence, thereby reducing EFHD2 transcription. Silencing FOXG1 suppressed EFHD2 and sensitized NSCLC cells to cisplatin The data is interesting. However, I suggest authors need to revise some major points, as listed below, to make this study complete.
1. The study didn’t contain any in vivo or clinical data to further confirm its reproducibility or authors could replenish a paragraph to discuss this.
2. Authors proposed inhibition of EFHD2/NOX4 pathway increase NSCLC cisplatin sensitization through ROS production, However, no result confirm the ROS level in cells. I suggest authors could replenish it.
3. The results of molecular docking better to be validated through co-immunoprecipitation (co-IP). Hence, I also suggest authors could replenish it.
4. The inhibitory activity of extracts is generally inferior to the pure compounds. Cause extracts contain multiple components, whereas purified compounds consist of a single active ingredient. However, author’s data showed that inhibitory effect of the extract (Fig 3) was better than the pure compound (Fig 4). I suggest that the authors could further discusses this phenomenon.
Comments on the Quality of English LanguageThe quality of English is acceptable.
Reviewer 4 Report
Comments and Suggestions for Authors
The article by Hu et al. demonstrates that a compound isolated from Stephania tetranda exerts positive effects on non-small cell lung carcinoma. Overall, the article is well-structured and clearly written. However, the authors should conduct additional experiments, address several issues, and improve the discussion of their results.
- Page 6, Lines 194-196: The authors state that they did not investigate argininosuccinate lyase because it is produced by the human body and therefore cannot be considered a drug. This statement is inaccurate, as there are numerous compounds, such as opioids and cardiac glycosides, that are used as drugs despite being naturally produced by humans.
- Cytotoxic Effects: The authors should present data on the cytotoxic effects of coclaurine on the cell lines used in this study.
- Figure 5B: The authors should repeat the experiments using higher concentrations of cisplatin and calculate how the IC50 is affected by cotreatment with coclaurine. Additionally, the authors should explain the rationale for using very high concentrations of coclaurine (200 μM), particularly since its toxicity is unknown.
- Cloning of the FOXG1 Gene Promoter: The Materials and Methods section lacks details about the cloning of the FOXG1 gene promoter. The authors should specify which fragment was cloned into which vector, and what was used as the control. Additionally, the activity of the empty control vector should be shown.
- Figure 6B: The qPCR results should be presented in absolute values with standard deviations for the controls.
- Thermal Shift Assay: The authors refer to a thermal shift assay, but they actually conducted a cellular thermal shift assay. This should be corrected, and the authors should indicate whether the western blotting was performed under reducing or non-reducing conditions.
- Molecular Docking: The molecular docking process needs clarification, particularly how it was performed on proteins that lack a ligand-binding domain.
- Bioavailability and Metabolism of Coclaurine: The authors should discuss the bioavailability of coclaurine and its potential metabolism. Is it feasible to achieve 200 μM concentration in human plasma? Does this compound bind to plasma proteins?
- Cisplatin Resistance Mechanisms: The authors should discuss the potential mechanisms of cisplatin resistance involving transcription factors in various cancers, citing relevant literature such as doi: 10.3390/biom12101365, doi: 10.2174/1568011053352587, doi: 10.1038/s41598-018-31030-3, and doi: 10.1016/j.ejphar.2023.175728.
Round 2
Reviewer 3 Report
Comments and Suggestions for Authors
The current from of manuscript is acceptable
Reviewer 4 Report
Comments and Suggestions for Authors
No further comments, the authors addressed all concerns.